# Hypnosis and Sedation for Anxious Children Undergoing Dental Treatment: A Retrospective Practice-Based Longitudinal Study

**DOI:** 10.3390/children9050611

**Published:** 2022-04-25

**Authors:** Sabine Rienhoff, Christian H. Splieth, Jacobus S. J. Veerkamp, Jan Rienhoff, Janneke B. Krikken, Guglielmo Campus, Thomas Gerhard Wolf

**Affiliations:** 1Pediatric Dental Practice, D-30177 Hannover, Germany; kontakt@magic-dental.de (S.R.); medizinproduktebeauftragte@magic-dental.de (J.R.); 2Department of Preventive and Pediatric Dentistry, Center for Oral Health, Ernst Moritz Arndt University Greifswald, D-17475 Greifswald, Germany; splieth@uni-greifswald.de; 3Kindertand, Pediatric Dental Practice, NL-1076 Amsterdam, The Netherlands; veerkampj@gmail.com; 4Snoet Kindermondzorgcentrum, Pediatric Dental Practice, NL-1061 Amsterdam, The Netherlands; jantjekrikken@gmail.com; 5Department of Restorative, Preventive and Pediatric Dentistry, School of Dental Medicine, University of Bern, CH-3010 Bern, Switzerland; thomas.wolf@unibe.ch; 6Department of Surgery, Microsurgery and Medicine Sciences, School of Dentistry, University of Sassari, I-07100 Sassari, Italy; 7Department of Periodontology and Operative Dentistry, University Medical Center of the Johannes Gutenberg-University Mainz, D-55131 Mainz, Germany

**Keywords:** compliance, hypnosis, midazolam, pediatric dental treatment, sedation

## Abstract

To assess whether the treatment of children with oral midazolam and pediatric hypnosis techniques can improve the compliance in consecutive sessions, a retrospective longitudinal practice-based observational study was designed and carried out. A total of 311 children between 3 and 12 years of age were treated under hypnosis and sedation with midazolam (0.40 mg/kg body weight). Treatments were performed in one to a maximum of three sessions. A total of 183 children received one, 103 received two and 25 children received three treatment sessions. The behavior of the children during the sessions was examined by means of the Venham score. The self-evaluation of the children was based on the Wong–Baker Scale. Child behavior using midazolam and hypnosis techniques showed little difference and good compliance between the sessions. Venham scores did not increase significantly regarding total treatment from the first (0.99 ± 1.41) to the second (1.17 ± 1.39) and to the third session (1.27 ± 1.20) (*p* > 0.05). However, considering the highest Venham scores that occurred in each case, the behavior of the children worsened significantly (*p* < 0.01) during the three treatment sessions, from 1.37 ± 1.31 (first) to 1.87 ± 1.74 (second) to 2.32 ± 1.33 (third). In 6.11% of the children, treatment was discontinued in the first session (*n* = 19), 0.96% in the second (*n* = 3) and 0% in the third. Treatment with low-dose midazolam, combined with hypnosis techniques, showed to be an effective option for dental treatment in children. Within the limitations of the current study, and with consideration of highest possible compliance, no more than two treatment sessions for pediatric dental treatment should be performed.

## 1. Introduction

The prevalence of dental fear is more than 25% in children and adolescents [1], and behavioral problems are a predictive factor for dental fear [2], usually leading to a delay or discontinuation of treatment [3]. However, fear does not necessarily cause uncooperative behavior at the dentist’s office [4,5]. The behavior of a child cannot be predicted by the anticipatory fear of the child itself or the parents. The skills of an experienced dentist can reduce the child’s fear of dentistry by using communication techniques [6]. In Germany, 65% of dentists, mainly male, feel stressed and overwhelmed when a young patient comes into the practice. They do not have the time to treat children, and the necessary training is often lacking [7]. Non-disabled children are still held or wrapped at the dentist’s chair, and treatment is carried out in this way [8]. While these previously common techniques, such as protective stabilization or the use of papoose boards, are still used in some countries, these practices are now banned, particularly in Scandinavian countries [9]. Verbal and pharmacological techniques are now the choice, and few parents agree to hold their child during dental treatment [10]. The most important object is rendering the first dental visit as pleasant and atraumatic as possible. The child should feel as though he or she has firm control over the procedure and that the treatment is predictable [11]. A step-by-step approach of gradual exposure, starting with a very simple treatment, which then increases from session to session and slowly shows the child the treatment steps, is a good technique to prevent dental fear [2,12]. Likewise, empathic positive communication with verbal explanations and combined physical contact with the child can promote the child’s cooperation. The combination of the two techniques has proven to be even more effective than practicing either one alone [13]. Hypnosis has been an established method for more than 40 years, not only for pediatric pain therapy, but now fully integrated in practices, clinics and hospitals [14]. As sole therapy or in combination with other techniques, hypnosis is used regularly and frequently in many clinical pediatric situations to change patients’ perception, thinking and behavior [15]; induce relaxation; and reduce anxiety and pain [11,16,17]. Sedation techniques are used in pediatric dental offices worldwide [18]. Compared to other drugs, midazolam might produce a particularly pronounced amnesia, especially regarding the memory of the local anesthesia [19]. Data on the dosage of midazolam vary in the literature [20,21,22], showing moderate-certainty evidence that oral midazolam is an effective sedative agent for more cooperative behavior of children during dental treatment [23].

Thus, the question arises whether midazolam can avoid conditioning the child to the dentist and how the child behaves when several consecutive sessions are required. It was shown that, with an increasing number of treatments of the same child, the behaviors change and deteriorate with each successive visit. This resulted in a maximum of two sedation sessions per child, treating as many teeth as possible per session and limiting the treatment to simple restorations and extractions [24].

The hypothesis behind this study was to clarify whether the treatment of children with oral midazolam and hypnotic techniques might improve the compliance of the children in up to three consecutive sessions. To assess the appropriateness of the hypothesis, a retrospective longitudinal practice-based observational study was designed and performed to evaluate the children’s behavior during pediatric dental treatment, using the Venham score [25] and Wong–Baker Scale [26].

## 2. Materials and Methods

### 2.1. Selection of Participants

The study was conducted in accordance with the principles for medical research involving human subjects described by the Helsinki Declaration. In this retrospective longitudinal observational practice-based study, children aged 3 to 12 years were considered eligible. The inclusion criteria were general medical healthy (ASA class I + II) with at least two treatments under sedation, anxious children with willingness to cooperate and dental treatment with restorative measures (fillings, pediatric crowns, pulpotomies and root-canal treatments) or extractions. Exclusion criteria were defined as serious general diseases ASA class ≥ III, age under 3 or over 12 years, only one treatment under sedation, treatment under general anesthesia, no sedation, no willingness to cooperate, respiratory-tract obstructions, severe overweight, weight under 10 kg, high extent of treatment and difficult surgical treatments. The study was conducted in a private dental pediatric practice (Hannover, Lower Saxony, Germany) by a single dentist specialized in pediatric dentistry and hypnosis. The dental treatments were carried out according to standardized treatment procedures and exclusively according to the individual needs of the child. No additional treatments were performed. In this specialized private dental practice, questionnaires are part of the standardized procedure of the dental treatment. Sedation is used on a routine basis, and its efficacy is continuously evaluated. All treatments were recorded with audio and video recordings. These videos were evaluated by two independent and previously calibrated dentists.

Sample size was calculated based on an appropriate sample size of previous studies for pediatric dental treatment of anxious children [2,5,27,28,29], with a power set at 80%. The previous sample size was increased by 20% (to 311 subjects) to ensure an optimal level of accuracy (5%), given the possible effects of dental anxiety and non-response.

### 2.2. Medical History

In the first treatment session, the child’s medical history was taken. This history included general medical conditions, medication, special dental history, diet, oral hygiene and preferences of the child. The anamnesis was taken both by means of an anamnesis sheet and in a personal conversation with the dentist. In the general medical anamnesis, questions were asked about organ diseases, previous operations or hospital stays, anesthesia and the intake of medication. Children must be fasting four hours before sedation in the participating practice. The parents/legal guardians were asked whether the child was developmentally delayed or had already had therapies such as speech therapy, occupational therapy or physiotherapy in order to determine the child’s developmental status and to find out whether the child should be classified in ASA classes I or II. During the dental anamnesis, the dentist asked about previous experiences at the dentist and the child’s behavior in this situation. The presence of radiographs and pain history of the children were asked. Furthermore, the reasons/concerns were asked, why they came into the pediatric dental practice and how they became aware of the specialized practice (e.g., referral by the pediatrician or family dentist, or own research). Questions were asked about the child’s diet, such as sugar consumption or the use of feeding or sports bottles. Suggestions for improvement and tips for changing the diet were given right at this meeting. The anamnesis about oral hygiene was also given, with corresponding information for parents and child.

Finally, a conversation with parents and child about the child’s preferences, such as favorite toys or hobbies, took place in order to establish contact with the child and to be able to interest the child in later treatment with stories or conversations about the favorite topics.

### 2.3. Recording of Findings and Treatment Planning

Following extra- and intraoral findings of the children at the first treatment appointment, radiographs were taken if indicated and possible. The child was then given a prophylaxis session by a dental assistant, during which brushing of the teeth was explained in a child-friendly manner, the teeth were brushed together with the child and age-appropriate nutritional tips were given. If possible and necessary, the dental assistant cleaned the teeth professionally with a polishing brush or cup and prophylaxis paste and fluoridated the teeth with fluoride varnish (Fluoridin, VOCO GmbH, Cuxhaven, Germany). The dentist discussed the treatment plan with the parents during the prophylaxis. First, the findings were explained to the parents, also based on the radiographs, and the treatment options were presented by using photos or models. The parents were informed about the advantages, disadvantages and risks of the treatment method. The risks include cardiovascular reactions, the possibility of respiratory depression and a paradoxical reaction. Written informed consent was obtained from the parents/legal guardians that the treatment of the child be carried out and recorded on video and used for study purposes. According to the legal basis prevailing in Germany, a treatment appointment could only be arranged after at least 24 h and the written informed consent of the parents. Finally, the child was called in again and the planned dental treatments were explained in a child-friendly manner. All children were given midazolam–ratiopharm^®^ 2 mg/mL oral solution (ratiopharm GmbH, Ulm, Germany) at a dose of 0.4 mg/kg body weight, considering the maximum value to prevent overdose (7.5 mg).

### 2.4. Preparatory Measures

At the treatment appointment, the child was taken by the receptionist to a quiet reception area. Oxygen saturation and heart rate were measured with a pulse oximeter, and the child was asked to indicate how he or she felt at that time, using the Wong–Baker Scale [26]. Then the oral-sedation juice was administered in a cup. To prevent overdose, no additional dose was given in any of the cases. The corresponding values (heart rate, oxygen saturation and amount of medication) were recorded on the monitoring sheet. The child was accompanied back to the waiting area and was allowed to choose a DVD from a list to watch during the treatment. During this time in the waiting room, the children are waiting for the treatment with the accompanying person(s) and an additional dental assistant who monitors the child until the time of treatment. The children or parents/guardians are required to accompany the child in the participating practice. If they come by car, even two accompanying persons must accompany, one person who drives and one person who monitors the child.

### 2.5. Treatment Goal and Treatment

Immediately before the treatment, it was planned which teeth were to be treated during each visit. As a rule, this was one quadrant per session, and in the case of small fillings that did not require anesthesia, if possible, one side of the jaw was treated. The treatment goal was individually adapted. If the child’s cooperation was very good, more teeth were treated than planned. If cooperation was poor, the treatment was shortened.

After an average waiting time of 25 min, the child was led to the treatment room. In general, the children in the performing practice are preferably led into the treatment room without their parents during treatments under sedation. If the parents did not give their consent, one parent was allowed to accompany the child. In the treatment room, the accompanying person (parent/legal person) was assigned a chair and instructed and asked to remain quiet and to be present as a silent observer, not interfering in the communication with the child. First, the child was placed on the treatment couch and was again to show on the Wong–Baker Scale [26] what his or her current condition was like. Second, he lay down on the couch and got a pulse oximeter on his finger, and then DVD-glasses (Zeiss Cinemiser OLED 3D, Carl Zeiss Meditec AG, Jena, Germany) with the selected film were put on. Even before the treatment, the children were told a hypnotic story. Most treatments were performed under local anesthesia with prior application of topical anesthetics. In the first session, there were 94 children (91%); in the second session, 93 children (90%); and in the 3rd session, all children received local anesthesia. Only in the case of a small flat filling, where no or very little pain was expected, local anesthesia was not used. The treatment was carried out as planned. All treatments were carried out under rubber dam if possible. All treatments were carried out by one dentist with two alternating dental assistants. All members of the dental team involved in the treatment were trained in behavioral management and hypnosis techniques and had at least 10 years of practical experience with children and sedation. During treatment, additional techniques from behavioral management, such as tell–show–do and hypnosis using double-induction techniques, were used as required by the child. In addition, care was taken to ensure that there was constant physical contact with the child by at least one hand of the practitioner or assistant. The hypnosis techniques used in the current study have been trained in the curriculum of the scientific society Deutsche Gesellschaft für Zahnärztliche Hypnose (German Society of Dental Hypnosis, DGZH e.V., Stuttgart, Germany), using a so-called force-animal induction, color induction, bird-swing induction, and magic-arm induction [30,31]. During the entire treatment, there was always a second dental assistant in the room. She concentrated on monitoring and recording the values of pulse and oxygen saturation. In addition, this dental assistant handed out necessary materials, such as cement mixed or selected children’s crowns, so that the dentist and chair assistant could fully concentrate on the treatment of the child. At the end of the treatment, the child was shown the Wong–Baker Scale [26] again and asked about his or her well-being. The treatment was recorded by a permanently installed video camera. The parents/legal guardians and child were then accompanied by a dental assistant to the recovery room. The parents were also informed about their behavior after the treatment. After 1 ½ h, the dentist checked whether the child could be discharged home. For this purpose, it was checked whether the child could give meaningful answers and walk alone without staggering. Afterward, the heart rate and oxygen saturation were measured again with a pulse oximeter, and the child was to show his or her condition on the Wong–Baker Scale [26] once again. If this was the case, the child was discharged home; if not, the child remained in the recovery room for a further period. Each further treatment session followed the same pattern, only the two questionnaires were not completed again. This was performed only at the first treatment session.

### 2.6. Evaluation of the Videos

The videos of treatment preparation and treatment were viewed by two independent calibrated dentists. To check intra- and inter-examiner reliability, the two independent dental examiners were trained before the start of the study. Forty subjects (age range 3 to 12 years) were examined and re-examined by the two independent examiners after 72 h. Using analysis of variance for fixed effects, inter-examiner reliability was assessed [32], whereas intra-examiner reproducibility was assessed as percent agreement and Cohen’s kappa statistic. Inter-examiner reliability was found to be good, with no significant differences (*p* = 0.22) and a low mean square error (0.40). The percent agreement in terms of reproducibility between investigators was high (Cohen’s Kappa 0.86). For the analysis of the video recordings, the examiners were blinded in terms of the number of sessions, i.e., they did not know whether there had been a treatment before. This was necessary to ensure objective observation of the treatment. The behavior of the child was determined at several points in time during treatment, using the Venham score [25]. This is a behavioral scale from 0 = absolutely cooperative child to 5 = absolutely uncooperative child. Before, during and after treatment, the Venham scores were collected at different points in time: local anesthesia, rubber dam, use of red handpiece with water, excavation with green handpiece, filling of the tooth, polishing, extraction and at the end of the treatment. Furthermore, the maintenance of a body contact by the dentist or the assistant was observed.

### 2.7. Statistics

The analyses were determined by using descriptive statistics. Two independent examiners observed scored all videos. Normally distributed continuous were data expressed as mean and standard deviation, and one-way ANOVA was used to look at the differences between the different groups/sessions. A Wilcoxon signed rank test was used to examine the comparability of the individual treatment sessions. The evaluation was performed by using SPSS Statistics 20 (IBM, Armonk, NY, USA).

## 3. Results

In this retrospective longitudinal observational study, a total of 311 children aged between 3 and 12 years were included over a period of 2 years. These were 142 girls and 169 boys with an average age of 74.22 months (SD ± 24.71, MIN 26 months, MAX 167 months). There was no significant gender-specific age difference between the two groups. The results of at least two consecutive treatment sessions were considered (Table 1 and Table 2).

This included 103 children, in whom a total of 235 treatments were performed under sedation with midazolam. The mean age was 68.64 months, and 44 (42:7%) of the children were girls. Twenty-five children had three treatments (of which, 11 were girls), and three children had four treatments (100% boys). Most of the children were accompanied by their parents; in only in a few cases (*n* = 7) were other legal guardians present. Usually, the child was accompanied by the mother (72%), and more rarely by the father (11%) or by both parents (17%). It is noticeable that the number of accompanying fathers dropped from 16% to 4% from the first to the third session (Table 2).

### 3.1. Behavior before and during Treatment

The child’s behavior was assessed by a second independent dentist based on the Venham score. Regarding the child’s behavior when drinking the midazolam, there was no difference at the first (0.49 ± 0.18) and second session (0.47 ± 1.23; *p* > 0.05). Overall, the values were very low and, thus, showed relaxed behavior. In the third treatment (*n* = 25), however, the Venham score was significantly lower, and the behavior was therefore better than in the second (0.05 ± 0.35; *p* = 0.02). Even directly before the treatment, there was no significant difference in behavior between the three treatment sessions (Table 3).

A clear statistically significant relationship was recorded between session number and the behavior of the children (*p* < 0.01). The highest value in each case was considered here. In the second treatment session, the children had a significantly higher Venham score (1.87 ± 1.74) and, thus, a worse cooperation than in the first session (1.37 ± 1.31). This trend also continued among the children who needed a third treatment session. The average value during the third session for these 25 children was 2.32 ± 1.33.

Table 3 shows the comparison of the mean values of the Venham score. When comparing the sessions in a Wilcoxon Signed Rank Test, in the comparison from first to second session only the children who had two sessions are considered (*n* = 103), and in the comparison from second to third session, only children who had three sessions are considered (*n* = 25). All other children are excluded from the analysis in Table 4 and Table 5.

In both cases (comparison of first and second session and comparison of second and third session), it is noticeable that the Venham score is higher and, therefore, the cooperation of the children in the following session is worse. In the comparison of the two Wilcoxon tests, however, the values for the comparison of the second and third session are lower than for the comparison of the first and second session (Figure 1).

In the comparison of the Venham score of the individual treatments, Figure 2 also shows that the cooperation of the children is lowest in the third session.

Only during polishing and when inserting the child’s crown are the values significantly lower than in the first and second session. The highest value of 1.52 is achieved in the third session when using the red handpiece. It is therefore even higher in the third session than the value for the extraction with 1.29. As these children were all anesthetized, the behavior can therefore be very uncooperative, even when using rotating instruments, and even exceed the value for the extraction.

### 3.2. Self-Assessment of the Children

The evaluation by an independent dentist is one way of assessing the children’s behavior and well-being during dental treatment. The second way chosen here is the self-assessment of the children by using the Wong–Baker Scale [26].

The children were asked at four points in time about their own well-being, using the Wong–Baker Scale (Figure 3).

At the time of drinking the midazolam juice, the children felt significantly better at the second session than at the first session (Z = −2.785, *p* < 0.01). Directly before and after the treatment and shortly before leaving the practice, there were no significant differences between the first and second treatment. At no time between the second and third treatment sessions were there significant differences in the self-evaluation of the children. Even though there were no significant differences between the second and third sessions, it is noticeable that the children felt worse in the third session than in the second when they were given juice. Overall, the graph shows that, in the first and second session, the children felt continuously worse from the time the juice was administered until the end of the treatment, but when they left the practice, they returned to the initial level of the first session. In the third session, it was worse at the beginning and improved again in course of the treatment.

## 4. Discussion

The aim of this retrospective longitudinal practice-based observational study was to assess the compliance of children under midazolam sedation with hypnotic techniques in consecutive treatment sessions, based on the Venham score and by self-assessment of the children. In order to obtain as independent an evaluation as possible, all treatments were recorded with a video camera to emotionally disconnect the observing dentist who was blinded regarding treatment sessions [33].

### 4.1. Behavior of the Children during the Treatment

It was found that the children’s cooperativeness decreased with each treatment session under midazolam and hypnotic techniques. In the present study, the dosage of 0.4 mg/kg body weight was used, derived from a study in preschool children, using 0.5 mg/kg body weight [20]. Data on the dosage of midazolam vary in the literature. A Cochrane review showed that there is moderate-certainty evidence that oral midazolam is an effective sedative agent for more cooperative behavior of children during dental treatment at a dosage between 0.25 and 0.75 mg/kg body weight [23]. It is also described that, the higher the dosage of the drug, the higher the cooperation of the child [22]. However, this is often also accompanied by a greater risk of side effects, especially paradoxical reactions [21], whereby a dose in the amount of 0.4 mg/kg body weight also rather prevents an overdose. The effects of the severity or extent of treatment on outcomes can be quantified mainly by the duration of treatment. It was observed in the present study that the effect of sedation was limited to about 25–30 min as an effective dose and was not exceeded. No significant difference was observed regarding the severity of treatment. For extractions, the compliance of children was lower compared to conservative treatments. Veerkamp et al. [27] observed that the peak of the Venham score increased slightly in consecutive sessions until the third treatment session. In a randomized control group without laughing gas, the behavior during treatment of consecutive sessions improved, but the Venham score was significantly higher than with laughing gas treatment, so that the behavior of the children with laughing gas was significantly better than without [27]. Kapur et al. [34] compared the Venham score in the treatment of children under midazolam sedation and with behavior without pharmacological help. The behavior is significantly better in the treatment with sedation than without midazolam. However, there are hardly any further works in the literature on this question.

From the data available, it can be concluded that sedation in anxious children is helpful to reduce anxiety levels or to improve behavior during treatment. However, learning success over several sessions does not occur to the desired extent. This could be due to the anterograde amnesia caused by the midazolam. Kain [35] observed that significant amnesia occurs when premedication is given ten minutes before treatment begins. This can last up to 48 h after drug administration [36]. If children are treated several times in routine situations, conditioning seems to take place, so that the children are more anxious and behave worse in the following session [37]. However, if the children have a good experience with dental treatment, in the sense of desensitization or model learning, collateral pathways appear to form in the brain, as is generally the case with learning [38]. This allows a new behavior to be learned. Because of the amnesia caused by midazolam, this positive learning does not seem to take place. At the same time, the midazolam may become accustomed to it, so that the dose would have to be increased in each session to achieve the same level of sedation; similar effects are found in intensive care [39] and in the general use of benzodiazepines, e.g., as a sleeping pill [40]. These results are consistent with those of Day et al. [24], who recommend limiting sedation with midazolam to a maximum of two sessions.

### 4.2. Self-Assessment of the Children

The Wong–Baker Scale was developed to assess pain [41]. Like comparable scales, it can also be used for preschool children from 3 years of age [42]. In this study, children were not explicitly asked about pain, but about their general mood. The study by Cravero et al. [26] shows that there is a high correlation between mood and pain, which can also be measured with the Wong–Baker Scale [43]. The only significant difference in the self-assessment of the children at the different sessions was that the children felt better at the second session when they were given the midazolam juice than at the first session. This appears to be an effect of positive conditioning [38], as the child was not yet able to develop amnesia at that time. Since the self-assessment did not show any significant differences at all other points in the interview, i.e., immediately before and after treatment and shortly before leaving the practice, it can be assumed that no negative conditioning took place, but rather a process of habituation to the midazolam seems to take place, meaning that the same depth of sedation is not achieved in consecutive sessions and, therefore, the behavior during treatment is worse.

It is interesting to note that the self-assessment of the children did not correspond to their behavior during the treatment, because at no time was there a significant difference in the self-assessment of the children before and after the treatment and when leaving the practice. While the highest value of the Venham score increases from session to session and, thus, the behavior, as well as probably also the child’s feelings, gets visibly worse, the children themselves do not indicate any difference in their condition after the treatment. It could be that they have forgotten a bad experience afterward or internalize it for themselves so that they do not show it to the outside world. Santamaria et al. [28] made similar observations in a study. The cooperation of the children was significantly different, but the self-assessment remained consistently good. This suggests that the children have developed externalized coping strategies for themselves, and this should be accepted by therapists. Children obviously show strong external reactions with which they process unpleasant feelings, but nevertheless evaluate the treatment positively. At this point, it should be mentioned that no quantifiable method was used to measure dental anxiety with a dental anxiety scale, and, thus, it is not known whether the children were anxious or what the reason was for not cooperating. One factor that can possibly influence the cooperation of the children is the accompanying person. Almost all children were accompanied by their parents to the treatment, and only 1% were accompanied by other persons, such as grandparents. The mother was most often present (Table 1). At the third treatment session, the father was much less often alone with the child in the practice, but, more often, both parents were present. Virdee and Rodd [44] have produced similar results. They described that 62% of the people coming for dental treatment were the mother, 13% both parents, 12% the father and 4% other people with the child.

### 4.3. Accompanying Measures through Behavioral Guidance and Hypnotic Techniques

In the literature, numerous techniques regarding behavioral management are described. These techniques vary greatly, depending on the dentist’s training and regional conditions, as well as cultural and philosophical influences. The most common are tell–show–do and a positive reward system [45]. In a survey in Israel, most dentists reported that they use the following techniques in descending order of frequency: (1) tell–show–do, (2) model learning, (3) voice control, (4) positive rewards, (5) parental restraint, (6) papoose board, (7) hand-over-mouth and (8) hypnosis [46]. A similar study in Norway showed a slightly different result: (1) tell–show–do, (2) relaxation techniques, (3) distraction techniques, (4) systematic behavior therapy and (5) sedation [47]. Such techniques often have the same success as, for example, sedation with nitrous oxide [48].

When parents are asked which techniques are preferred, it is mainly the less restrictive measures that they would like to see for their children. Spanish parents indicated tell–show–do as their preferred technique but did not accept “hand-over-mouth” and the use of the Papoose board. For other accompanying measures, there were differences in preference according to the socioeconomic status and gender of the parents [49]. The same result was found for British [50] and Saudi Arabian parents. These parents also stated that a reward system and distraction were positive, while the separation from the child and “voice control” were negative [51].

Davies and Buchanan [52] asked the children about this topic and again got different results. The following techniques were accepted by the children in descending order: (1) stop signals, (2) distraction, (3) communication, (4) positive reward, (5) tell–show–do, (6) (“sensation information”) information about feelings, (7) inhalation sedation and (8) voice control. Restrictive measures were not queried in this study but would certainly have been in last place.

What they all have in common is that a gentle, positive, calm, explanatory behavior on a verbal level is desired, which, if possible, does not exert any coercion on the child. Children also wish to be able to interrupt the treatment at any time.

In the present study, techniques from hypnosis and tell–show–do were used. Unfortunately, there are hardly any works in the literature available on the effect of hypnosis in pediatric dentistry. According to a Cochrane review, there is not enough evidence to consider the benefits of hypnosis as established [53]. Nevertheless, there are some authors who describe the positive effects of hypnosis in children in dentistry [54]. For example, the visualization of a memory or an image during the administration of local anesthetics can be helpful [55], and even the pulse rate and crying of the child could be reduced in this way compared to a control group [56]. A form that is also frequently used in this study is the hypnotic confusion technique [57]. When looking at the results of the accompanying therapy, it can be observed that, in all treatments, constant physical contact with the child was held by the hand on the shoulder. This is not, as with the described “Restraint-Forms”, a holding of the child, but only a gentle, empathic touching of the child. For example, physical contact is one of the most common methods used by mothers to soothe a crying baby [58]. Guéguen et al. [59] have shown that a light touch by the doctor to the patient increases compliance, and the patient feels that the doctor is more concerned about the patient. The body contact and confusion techniques we used were described in detail by Schoderböck [60]. The child thus feels, all the time during the treatment, that he or she is being accompanied and not left alone at any time. The use of this technique was maintained throughout the entire study.

The verbal techniques varied according to the time of treatment. Thus, from the very beginning, verbal contact with the child was established, and a story was told alternately by the dental assistant and the dentist, using the confusing technique. It is noticeable that this technique was changed when the green dental handpiece was used. Here, the noise and the vibrating feeling of this device is imitated from both sides (dental assistant and dentist) by the talk-together technique to accompany the child and the sensations that arise. The tell–show–do technique was mainly used before the start of treatment with rotating instruments to explain the devices to the child. Overall, however, this measure was rarely used. In the third session the tell–show–do technique was significantly reduced compared to the first session, because it was assumed that the child was already familiar with the devices and that a further demonstration was no longer necessary. Sporadically, the counting technique was also used with rotating instruments. This was always combined with a short break, i.e., used in the sense of “positive reinforcement”. It should be considered within the limitations of the current study that numerous variables, such as constant physical contact with the child by at least one hand of the practitioner or assistant, that some children did not require anesthesia, type of dental treatment and accompanying persons were not considered in particular regarding the effect on the results.

In summary, it can be stated that the children were also very well-conditioned with verbal techniques. These are used to distract, explain to and motivate the patient. Although they are used intuitively and according to the needs of the individual child, they show a certain pattern. The verbal techniques correspond to the wishes of parents and children, as indicated in other studies. Only a stop signal was practically not agreed to with the children. This could be incorporated into the treatment in the future. All the techniques given can only be presented descriptively. They are used very individually, according to the respective needs of the child, and can hardly be standardized; therefore, they are difficult to evaluate. Nevertheless, a clear pattern of application has been shown. The hypnosis techniques require thorough training and a well-functioning treatment team. A very decisive technique seems to be the constant body contact. It is a non-verbal instrument that can be used in the same way for all children. The child does not yet need a cognitive understanding of spoken words, as it is not yet present in very young children anyway. It is also very well suited for children who cannot understand language. Through touch, the practitioner simultaneously receives feedback on the child’s condition. In this way, small movements that may express discomfort or pain can be felt immediately. It is also possible to have direct control of the heartbeat. This can help the dentist to react very quickly to the child’s needs. The touches can also help the child to control his or her own behavior. Gentle pressure on the shoulder indicates that the child should lie still, or a finger between the lower lip and chin can indicate that the mouth should remain open. The verbal hypnosis techniques, on the other hand, must be used very specifically. They are adapted to the age and sex of the child and the respective treatment step. In this way, the child is dissociated from the treatment or feelings during the treatment, and it can be explained by familiar everyday things. For example, the humming, vibrating noise when using the green dental handpiece with a tractor can be put into a different context.

A good concept of behavioral guidance or accompaniment by hypnosis may be beneficial for all treatments with midazolam, because considerable advantages for the patient result. These techniques used must certainly be applied very individually, depending on the needs of the child and the presentation or training of the dental team.

## 5. Conclusions

Within the limitations of the current study, it can be concluded that hypnosis techniques combined with sedation in anxious children are effective and helpful to reduce the anxiety level and to improve compliant children’s behavior during dental treatment. The study also shows that, in combination with hypnosis, sedation with a very low dose of midazolam (0.4 mg/kg body weight) may be successfully used in anxious children. As the children’s behavior deteriorated over the course of several treatment sessions, treatment under midazolam sedation should be limited to two sessions, if possible. The techniques used should be applied at the right time of treatment, which is sometimes hard to find and should be individually adapted to the dentist’s skills and the patient’s needs. Explanations for the child are essential to be given regularly. Further research is needed using a prospective clinical setting.

## Figures and Tables

**Figure 1 children-09-00611-f001:**
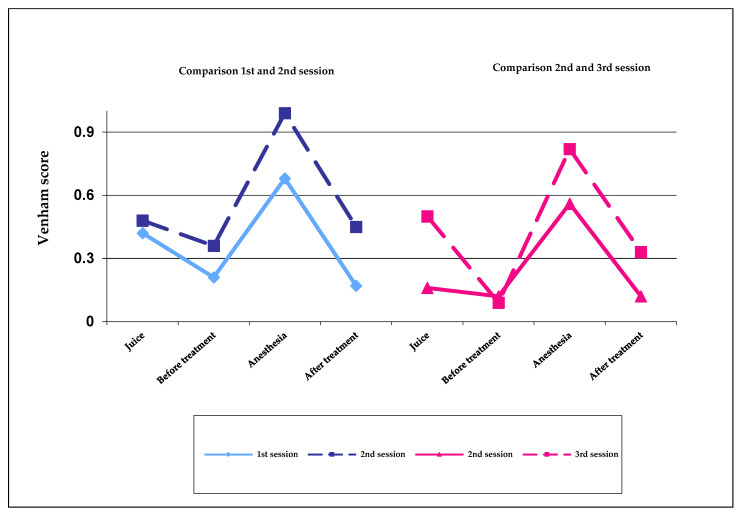
Wilcoxon test to compare treatment sessions.

**Figure 2 children-09-00611-f002:**
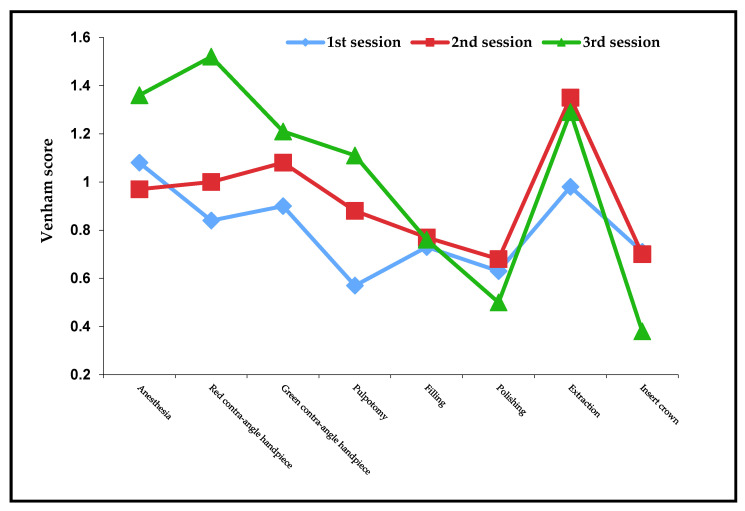
Venham score of the individual treatments.

**Figure 3 children-09-00611-f003:**
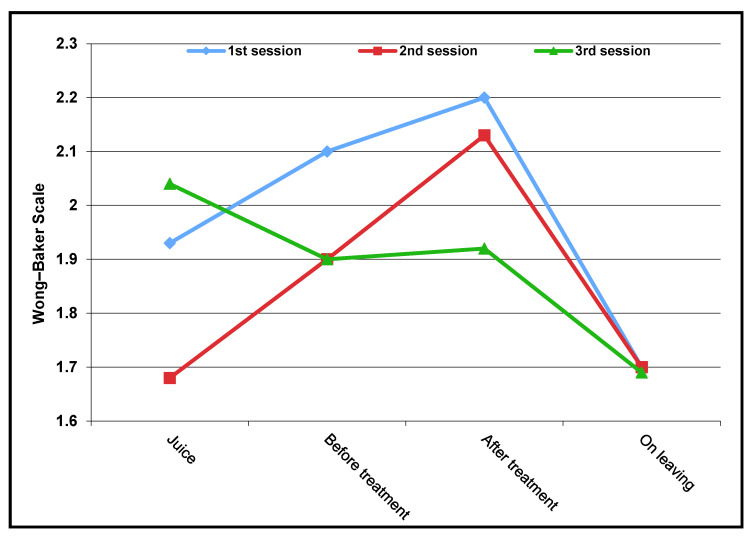
Self-assessment of the children during treatment.

**Table 1 children-09-00611-t001:** Type of dental treatment of the 1st, 2nd or 3rd treatment session, with number of children (*n*) and percentage (%).

Code	Treatment	1st Session	2nd Session	3rd Session
		*n* Children (%)	*n* Children (%)	*n* Children (%)
**0**	No treatment	17 (5.5)	4 (3.9)	0 (0.0)
**1**	Restoration/strip crown without anesthesia	38 (12.2)	6 (5.8)	0 (0.0)
**2**	Restoration and/or strip crown with anesthesia	36 (11.6)	20 (19.4)	5 (20.0)
**3**	Extraction	62 (19.8)	14 (13.6)	4 (16.0)
**4**	Extraction and restoration	38 (12.2)	10 (9.7)	3 (12.0)
**5**	Steel crown with pulpotomy or root treatment	81 (26.0)	40 (38.8)	8 (32.0)
**6**	Pulpotomy or root canal treatment without steel crown	13 (4.2)	1 (1.0)	2 (8.0)
**7**	Extraction and steel crown with pulpotomy	9 (2.9)	2 (1.9)	0 (0.0)
**8**	Miscellaneous	4 (1.3)	0 (0.0)	0 (0.0)
**9**	Steel crown	13 (4.2)	6 (5.8)	3 (12.0)
	Total treatments	311 (100.0)	103 (100.0)	25 (100.0)

1st session versus 2nd session Fisher’s exact test *p* < 0.01.

**Table 2 children-09-00611-t002:** Type of accompanying persons of the 1st, 2nd or 3rd treatment session with number of children (*n*) and percentage (*%*).

Session	Mother	Father	Both Parents	Others
Treatments	*n* Children (%)	*n* Children (%)	*n* Children (%)	*n* Children (%)
**1st session**	209 (69.0)	50 (76.9)	47 (73.4)	3 (60.0)
**2nd session**	75 (24.7)	14 (21.5)	12 (18.8)	2 (40.0)
**3rd session**	19 (6.3)	1 (1.6)	5 (7.8)	0 (0.0)
**Total treatments**	303 (100.0)	65 (100.0)	64 (100.0)	5 (100.0)

**Table 3 children-09-00611-t003:** Venham scores (intra-patient; 1st session *n* = 183; 2nd session *n* = 103; 3rd session *n* = 25).

(Mean/±SD)	1st Session	2nd Session	3rd Session	*p*-Value *
**Juice**	0.49 ± 1.18	0.47 ± 1.23	0.05 ± 0.35	NS/0.02
**Before treatment**	0.23 ± 0.89	0.35 ± 0.91	0.24 ± 0.66	0.03/0.03
**Anesthesia**	1.08 ± 1.44	0.96 ± 1.34	1.36 ± 1.38	NS/0.04
**Treatment**	0.99 ± 1.41	1.17 ± 1.39	1.27 ± 1.20	NS/NS
**End of treatment**	0.44 ± 1.10	0.44 ± 0.91	0.01 ± 0.14	NS/<0.01

* The first *p*-value refers to the comparison among 1st and 2nd session; the second *p*-value on the comparison among 2nd and 3rd session.

**Table 4 children-09-00611-t004:** Comparison of Venham scores in 1st and 2nd session (*n* = 103).

	1st Session	2nd Session	*p*-Value
**Juice**	0.42 ± 1.12	0.48 ± 1.25	NS
**Before treatment**	0.21 ± 0.89	0.36 ± 0.93	NS
**Anesthesia**	0.68 ± 1.17	0.99 ± 1.37	0.04
**After treatment**	1.17 ± 0.59	0.45 ± 0.92	<0.01

**Table 5 children-09-00611-t005:** Comparison of Venham scores of the 1st and 3rd session (*n* = 25).

	2nd Session	3rd Session	*p*-Value
**Juice**	0.16 ± 0.47	0.50 ± 1.41	0.01
**Before treatment**	0.12 ± 0.60	0.09 ± 0.29	NS
**Anesthesia**	0.56 ± 0.92	0.82 ± 1.26	NS
**After treatment**	0.12 ± 0.44	0.33 ± 0.76	0.04

## Data Availability

The data analyzed in this study are subject to the following licenses/restrictions: General European Data Protection Regulation (GDPR) of 25 May 2018. Requests to access these datasets should be directed to the corresponding author.

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
