# Peer review of "Hypnosis and Sedation for Anxious Children Undergoing Dental Treatment: A Retrospective Practice-Based Longitudinal Study"

_children, 2022, doi:10.3390/children9050611_

Round 1
Reviewer 1 Report
Article has flaws, additional experiments needed, research not conducted correctly.
There is evidence from numerous articles demonstrating that dental fear is not minimized by sedation or hypnosis techniques. It has been shown that regular dental check-ups, age at first visit, non-aversive visits support the phenomenon of latent inhibition, sedation does not. In addition, the way to avoid dental fear or to minimize it and achieve adequate collaboration is to confront the problem, in order to develop the necessary coping skills. In spite of the medical history reported in the manuscript, since we do not use a quantifiable method of measuring dental fear with a dental fear scale we do not know if the children had it or what was the reason why they did not collaborate.
Reviewer 2 Report
Hypnosis and sedation for anxious children undergoing dental treatment: A retrospective practice-based observational study
TITLE:
The manuscript title is appropriate and likely to serve its purpose.
The author did not submit a short title.
Keywords have been suggested.
ABSTRACT: An unstructured abstract is included.
INTRODUCTION/LITERATURE REVIEW:
The author cited the pertinent literature to some extent but needs more literature about low doses of midazolam.
The purpose of the paper was not stated clearly.
Suggested references to add to introduction and discussion:
- Jain SA, Rathi N, Thosar N, Baliga S. Midazolam use in pediatric dentistry: a review. J Dent Anesth Pain Med. 2020 Feb;20(1):1-8.
- Ashley PF, Chaudhary M, Lourenço-Matharu L. Sedation of children undergoing dental treatment. Cochrane Database Syst Rev. 2018 Dec 17;12(12):CD003877.
- Gentz R, Casamassimo P, Amini H, Claman D, Smiley M. Safety and Efficacy of 3 Pediatric Midazolam Moderate Sedation Regimens. Anesth Prog. 2017 Summer;64(2):66-72.
METHODS:
No control group was used.
Please mention the power of the sample size used in this study.
The authors stated [The child was 158 accompanied back to the waiting area}. How the child was monitored after giving the sedative?
The authors did not mention about the fasting of children which is a must before sedation.
Too many variables such as there was constant physical contact with 188 the child by at least one hand of the practitioner or assistant, some children did not require anesthesia, type of dental treatment, accompanying persons, etc. How that affect the results. It should be included in the discussion.
The authors stated [And in the 3rd and 4th sessions children received local anesthesia]. Nothing mentioned in the abstract, methods, purpose, or results about 4th session.
The author is not clear about the hypnosis methods used. He mentioned the children were told a hypnotic story and he used double induction techniques. These should be detailed and referenced with appropriate reference.
RESULTS:
Table 1 show 101 children for visit 2 while it is 103 else where.
Venham scores was not recorded during treatment in Figure 2 which may be critical.
The children were asked at four points in time about their own well-being using the Wong Baker scale. Two of these four were before and after treatment which the child may be under the sedative effect and their response may be questionable. Please elaborate.
TABLES AND FIGURES:
Table 1 is too crowded with the content and should be divided into two.
DISCUSSION:
The authors need to be sure that any new results discussed have previously been presented in the Results section.
Effect of 1 vs. 2 vs. 3 times. Please elaborate.
Effect of severity or extensiveness of treatment on the results. Please elaborate.
Too many variables such as there was constant physical contact with 188 the child by at least one hand of the practitioner or assistant, some children did not require anesthesia, type of dental treatment, accompanying persons, etc. How that affect the results. It should be included in the discussion.
Venham scores was not recorded during treatment in Figure 2 which may be critical.
The children were asked at four points in time about their own well-being using the Wong Baker scale. Two of these four were before and after treatment which the child may be under the sedative effect and their response may be questionable. Please elaborate.
CONCLUSIONS:
Part of the conclusion is recommendation and should not be in the conclusions. Please revise.
Round 2
Reviewer 1 Report
The authors have made a great effort to improve the manuscript with the required changes.
Reviewer 2 Report
Authors did address the reviewer comments.
Thank you